# Can good ESG performance of listed companies reduce abnormal stock price volatility? Mediation effects based on investor attention

**Fengju Wu**[1]*, **Bao Zhu**[2], **Siqi Tao**[1]

**1** Nanjing Audit University Jinshen College, Nanjing, China, **2** School of Social Audit, Nanjing Audit University, Nanjing, China

* wufengju@naujsc.edu.cn

**Data Availability Statement:** All relevant data are within the paper and its Supporting Information files.

**Funding:** The author(s) received no specific funding for this work.

## Abstract

Today, with a growing emphasis on sustainable economic development, corporate environmental, social and governance (ESG) performance is attracting increasing attention and favor from investors. This triggers a question: can good ESG performance of listed companies mitigate the "up and down" of the stock market by drawing investor attention? This paper utilizes the data from China's A-share listed companies from 2011 to 2020, with investor attention as a mediating variable, to explore how the ESG performance of listed companies influences abnormal stock price volatility. The findings suggest that stronger ESG performance of listed companies significantly reduces abnormal stock price volatility, in which investor attention plays a partial mediating role. This paper confirms the robustness of the findings through multiple robustness and endogeneity tests. Heterogeneity analysis reveals that listed companies with good ESG performance during the growth period are more likely to significantly mitigate abnormal stock price volatility. Similarly, firms that maintain commendable ESG performance in bear markets significantly reduce abnormal stock price volatility. These findings enrich the theoretical research on the impact of ESG performance on abnormal stock price volatility, provide empirical evidence for listed companies to emphasize ESG investment and encourage investors to consider ESG ratings. Additionally, the study provides a new perspective for government agencies to utilize corporate ESG performance to maintain the sound development of the capital market.

## 1 Introduction

In recent years, environmental issues such as global resource misuse, pollution and climate change have become increasingly serious, while social problems such as the wealth gap and public health crises have become more prominent, which has created an existential and developmental challenge for mankind [1]. To address this crisis and promote sustainable development, the United Nations Environment Program (UNEP) first introduced ESG in 2004. ESG,

**Competing interests:** The authors have declared that no competing interests exist.

which stands for Environment, Social and Governance, urges companies to balance economic interests and social benefits. ESG has now been widely recognized around the world. In 2020, China put forward its dual carbon goals of "carbon peaking" and "carbon neutrality." In the U.S., the Inflation Reduction Act of 2022 allocated $369 Billion for energy security and climate transition investments. In 2023, the European Union (EU) released the "Green New Deal Industry Plan for the Net Zero Era", and South Korea implemented its "Green Growth" strategy to revitalize its economy. Corporate ESG performance has thus become a major concern for both the practical and academic worlds.

According to risk management theory, ESG practices constitute an effective risk management tool. In the environmental (E) domain, adherence to sound environmental practices can significantly reduce the risk of legal actions and financial losses that may arise from non-compliance with environmental regulations. In the social (S) dimension, active fulfilment of social responsibilities, such as ensuring equitable working conditions and engaging in charitable activities, stabilizes the workforce and mitigates reputation damage due to adverse social impacts. In terms of governance (G), a standardized and effective governance framework can prevent internal fraudulent activities, enhance decision-making efficiency and transparency, and reduce the risk of regulatory sanctions due to poor governance. In summary, strong ESG performance indicates a company's heightened focus on environmental protection, social responsibility, and corporate governance throughout its operations, thereby lowering potential legal, reputational, and regulatory risks as well as the likelihood of abnormal stock price volatility. Existing literature mostly studies how to reduce abnormal stock price volatility from the perspective of corporate governance, while relatively few study the impact on abnormal stock price volatility from an integrated ESG perspective. Moreover, there are significant differences in the findings. Some scholars argue that corporate ESG performance significantly dampens abnormal stock price volatility, with innovation efficiency mediating the relationship [2]. Yet another study found a significant positive relationship between corporate social responsibility performance and stock price volatility [3]. It is also suggested that corporate ESG performance does not have a significant impact on the risk of stock price crashes, especially in countries with weak investor protection and regulatory enforcement [4]. The existing literature has not developed a unified conclusion on the relationship between corporate ESG performance and abnormal stock price volatility. This study aims to address the existing gaps in the literature by examining the relationship between ESG performance and abnormal stock price volatility from an integrated perspective and by exploring the mediating role of investor concern. By doing so, this study contributes to a deeper understanding of the mechanisms underlying the impact of ESG performance on stock price volatility, providing valuable insights for both academia and practitioners.

According to signaling theory and reputation theory, good ESG performance by listed companies sends a positive signal to the outside world that the company is committed to environmental and social responsibility and has a strong capacity for sustainable development in the future. Such companies usually have a better social reputation and are more likely to attract the attention and trust of investors. Meanwhile, according to the theory of bounded rationality, investors' decisions are often influenced by emotions, biases and limited information processing ability. Listed companies with good ESG performance are more likely to stimulate investors' moral and emotional considerations, thus drawing greater attention and investment inclination. According to the International Sustainable Investment Alliance, more than half of European investors consider ESG rating indicators when making asset investment decisions [5]. Similarly, institutional investors in China's A-share market show a clear preference for listed companies with good ESG performance [6]. Some scholars have found that an increase in investor attention significantly exacerbates the risk of a stock price crash in the next period,

referred to as the "crash effect of attention" [7]. Normal stock price volatility can promote optimal resource allocation and ensure the healthy development of the capital market [8]. However, abnormal stock price volatility increases investor risk, weakens the ability of arbitrageurs to correct market deviations, and raises the financing costs for listed companies [9]. This scenario is detrimental to the resource allocation utility of the capital market and the development of the real economy. Amid a sluggish global economic recovery, stock price volatility has intensified in various countries, and the Chinese stock market has long suffered "ups and downs", so it is both urgent and necessary to study how to reduce the abnormal stock price volatility to promote sound enterprise development.

Although some existing literature has found that good ESG performance by companies can reduce abnormal stock price volatility, both theory and practice suggest that good ESG performance attracts more investor attention. Increased investor attention, in turn, may exacerbate abnormal stock price volatility. This raises a question: does good ESG performance by listed companies reduce abnormal stock price volatility by increasing investor attention? What mediating effect does investor attention play? Is this relationship heterogeneous across different stages of firms' life cycles or varying market conditions? Clarifying these issues can help guide investors to hold shares of companies with good ESG performance for the long term, curb abnormal stock price volatility, and protect the health and stability of the capital market; on the other hand, it can motivate listed companies to emphasize ESG investment, promote the coordination of economic and social values, and boost the high-quality economic development. Based on a unique dataset of listed companies in China, this study empirically investigates the relationship between ESG performance and abnormal stock price volatility. It also examines the mediating role of investor concerns. The aim is to fill a gap in the existing literature. By doing so, this study seeks to provide theoretical guidance and practical reference for investors, corporate decision-makers and policymakers.

The existing literature has not reached a unified conclusion on whether good corporate ESG performance reduces abnormal stock price volatility, and there is almost no literature on the mechanism of investor attention in the relationship between ESG performance and abnormal stock price volatility. In view of this, this paper empirically investigates the relationship between ESG performance and abnormal stock price volatility based on Huazheng ESG rating data and the stock price volatility data of Chinese A-share listed companies in Shanghai and Shenzhen from 2011 to 2020. It explores the mediating role of investor attention and analyzes heterogeneity in the context of the corporate life cycle and bull and bear market conditions. Compared with previous empirical studies, the uniqueness of this study lies in the use of investor attention as a mediating variable to reveal whether good ESG performance of listed companies can reduce abnormal stock price volatility. This study seeks to provide empirical evidence for investors in-stock selection and for corporate decision-makers regarding ESG investment. It was found that good ESG performance of listed companies significantly reduces abnormal stock price volatility, with investor attention as a particular mediator. Further research reveals that good ESG performance of listed companies in their growth period can more significantly suppress abnormal stock price volatility. Besides, good ESG performance of listed companies during the bear market conditions significantly reduces abnormal stock price volatility.

The potential contributions of this paper are as follows: first, from the perspective of investor attention, this study reveals the internal mechanism by which listed companies' ESG performance affects abnormal stock price volatility. It confirms that investor attention plays a positive mediating role, thus bridging a gap in the existing literature; second, this study enriches the research on the economic consequences of corporate ESG performance and the factors influencing stock price volatility. It verifies that good ESG performance of listed companies can inhibit abnormal stock price volatility, providing evidence for the government to

incentivize corporate executives to focus on ESG investments and promote sustainable economic development; third, the heterogeneity analysis conducted in this study provides valuable empirical data for listed companies to examine corporate ESG value under different life cycles and capital market conditions. This analysis provides new perspectives for securities regulators to promote the healthy development of the capital market.

The subsequent sections of the paper are structured as follows: Section 2: Literature Review; Section 3: Theoretical Analysis and Research Hypotheses; Section 4: Research Design; Section 5: Empirical Analysis; Section 6: Heterogeneity Analysis; and Section 7: Research Conclusions and Recommendations.

## 2 Literature review

Regarding the relationship between firms' ESG performance and abnormal stock price volatility, the existing literature mostly focuses on the governance (G) perspective. Scholars have found that increasing incentives for executive compensation [10], the proportion of shares held by external institutional investors [11], and corporate information transparency [9] significantly reduce abnormal stock price volatility while increasing the proportion of firms' equity pledges exacerbates stock price volatility [12]. From the environmental (E) perspective, some studies indicate that green investment expansion significantly reduces stock price volatility [13], but other scholars believe that good environmental performance (E) does not necessarily reduce the risk of stock price collapse [14]. From the social (S) perspective, some scholars have found that there is a market premium for companies with strong social responsibility in a stock market crash, which can increase investor identification and stabilize stock prices [15]. Besides, the higher the quality of CSR disclosure, the lower the abnormal stock price volatility [16]. However, some scholars have found a positive correlation between CSR activities and stock price volatility [3]. From the perspective of comprehensive ESG performance, some scholars argue that corporate ESG performance can reduce information uncertainty and suppress abnormal stock price volatility through the "noise reduction effect" [17, 18]. Corporate ESG disclosure is believed to reduce stock price volatility through the information effect, the internal governance effect and the external reputation effect [19, 20]. However, some scholars contend that ESG performance has no significant effect on abnormal stock price volatility [21]. It can be seen that there is a wide divergence in the existing literature regarding whether corporate ESG performance reduces abnormal stock price volatility.

Scholars generally agree that listed companies with good ESG performance attract more investor attention [19, 22]. However, the impact of investor attention on stock price volatility is debated between two schools of thought: the "deterioration theory" and the "stabilization theory". According to the "deterioration theory", the attention of retail investors to individual stocks will drive investor trading [23], significantly increasing stock liquidity. The increased turnover rate will attract more short-term speculators who contribute to a "herding effect," leading to an increased stock price volatility. In contrast, the "stability theory" analyzes the situation from the perspective of information asymmetry. It argues that increased attention from retail investors promotes corporate information disclosure [24], thus reducing irrational operation behavior caused by misinformation. The increase in institutional investors' attention to a company can further reduce the irrational noise trading of retail investors, thus stabilizing the stock price. Since good ESG performance of listed companies tends to attract more investor attention, a question arises: Does increased investor attention exacerbate or mitigate stock price volatility? This question has hardly been studied in the recent literature. This paper aims to fill the research gaps through an in-depth study of this topic.

In short, although literature indicates that increased green investment, social responsibility or good ESG performance can significantly dampen stock price volatility, others argue for a positive or insignificant effect, and thus the existing literature does not reach a consistent conclusion. Meanwhile, there is a gap in the literature regarding the mediating effect of investor attention in the relationship between ESG performance and abnormal stock price volatility. Scholars have formed two distinct schools of thought on the impact of investor attention on stock price volatility. The differences and gaps hinder effective guidance for companies and investors to explore the value of ESG practices and curb abnormal stock price volatility in the capital market. In response, this paper empirically investigates the impact of corporate ESG performance on abnormal stock price volatility using data from Chinese A-share listed companies in Shanghai and Shenzhen. It analyzes how investor attention mediates this relationship. The aim is to provide empirical data for investors' stock selection decisions, companies' emphasis on ESG investments, and governmental efforts to promote the healthy development of the capital market.

# 3 Theoretical analysis and research hypotheses

## 3.1 ESG performance and share price volatility

According to Resource Dependence Theory, an organization's behavior and decision-making processes are significantly influenced by the external environmental resources it depends on. To survive and thrive, businesses must interact with other organizations in the external environment to acquire and safeguard these crucial resources. Companies that prioritize ESG performance effectively communicate their commitment to environmental stewardship and social responsibility. This helps them garner government and social support, and establish strategic alliances, mitigate resource constraints in their operations [25], thereby reducing business risks and abnormal stock price volatility. Specifically, enterprises emphasizing ESG performance tend to proactively purchase energy-saving and emission-reducing equipment and increase efforts in technological innovation to achieve green transformation, which minimizes inefficient investments, enhances investment efficiency [26], secures government support, improves production efficiency, strengthens sustainable competitive advantages, and lowers operational risks. As a result, it reduces abnormal stock price volatility. Moreover, enterprises that actively assume environmental and social responsibilities are more likely to obtain high-quality resources and market opportunities. They enjoy various support policies such as government subsidies, tax incentives, and bank credit [27], all of which ease financing constraints and debt pressures, preventing abnormal stock price volatility caused by insolvency. In summary, by enhancing ESG performance, enterprises can gain greater control and autonomy in a resource-dependent environment. They effectively reduce risks associated with external interactions, maintain organizational stability and minimize abnormal stock price volatility.

According to Stakeholder Theory, a corporation serves as a community that connects various stakeholders. It holds accountability not only to shareholders but also to other stakeholders including governments, employees, customers, and communities. Good ESG practices facilitate effective communication and mutual cooperation between corporations and their stakeholders, enhancing stakeholder recognition and establishing long-term, cooperative relationships [28]. From an internal governance perspective, a standardized and orderly governance structure ensures fairness among stakeholders, preventing major shareholders or management from power abuse to manipulate stock prices. This governance stability reduces the risk of abnormal volatility. From an external perspective, considering long-term win-win cooperation, external stakeholders often expect listed companies to pay attention to corporate

ESG ratings. Such external pressure compels companies to lay greater emphasis on environmental protection, actively fulfill social responsibilities, improve internal governance, and reduce speculative investment behaviors [29]. They are encouraged to make moderate and efficiency-oriented investments [6]. Ultimately, this approach helps stock prices better reflect the company's intrinsic value and reduces abnormal price volatility. In summary, by integrating the expectations and needs of both internal and external stakeholders and implementing high-standard ESG practices, corporations not only help build harmonious stakeholder relationships but also stabilize stock prices and minimize abnormal volatility.

Based on the above analysis, this paper proposes the first research hypothesis:

H1: Good ESG performance of listed companies can significantly reduce abnormal stock price volatility.

### 3.2 ESG performance, investor attention and abnormal stock price volatility

According to Signal Theory, companies with excellent ESG performance often exhibit a stronger willingness to disclose information [30]. This disclosure includes annual reports, environmental reports, and social responsibility reports, which convey positive signals such as social responsibility, orderly internal governance, and strong sustainable development capabilities [31]. These signals attract increased investor attention. From the perspective of Reputation Theory, companies with good ESG performance exhibit a strong sense of social responsibility and maintain a standardized corporate governance mechanism, enjoying a high external social reputation. A good reputation not only improves corporate financial performance and financing capabilities but also mitigates operational risks during crises [32]. As a result, these companies receive greater favor and attention from external investors. Additionally, companies with good ESG performance receive more praise from securities analysts for their good social reputation [5]. Professional analysts interpret company information, acting as information intermediaries to convey it to investors in the capital market [33]. It is evident that compared to listed companies with poorer ESG practices, those with good ESG performance enjoy a higher social reputation, making them more attractive to investor attention.

According to the theory of information asymmetry, there is a pervasive issue of information asymmetry between corporations and external investors. Rumors stemming from this asymmetry can trigger excessive trading behaviors among investors, leading to abnormal stock price volatility. For listed companies with good ESG performance, a high-quality information disclosure mechanism can effectively mitigate this information asymmetry, thereby enhancing corporate information transparency and curbing extreme investor sentiment [34]. This transparency allows investors to accurately assess the expected returns and risks related to corporate stocks, thus increasing rational investments and preventing drastic stock price volatility [35]. Consequently, stock prices are driven closer to their corporate value [22]. The better the ESG performance, the more optimism and attention it takes from investors [36], accelerating the incorporation of information into stock prices, reducing financial and systemic risks, earnings drift, and thus abnormal stock price volatility. According to the theories of bounded rationality and behavioral finance, investors have limited attention when faced with a vast amount of information and a limited range of decision alternatives. The limited focus of different investors leads to varying beliefs about stock prices among market participants, resulting in frequent trading behaviors and consequent stock price volatility. However, listed companies with excellent ESG performance rely on their good social image and reputation to quickly capture the limited attention of investors. This enhances investor confidence, soothes investor unease [37,

38], and reduces irrational stock trading behaviors, thereby exerting a suppressive effect on abnormal stock price volatility.

Based on the above analysis, this paper proposes the following research hypotheses:

H2: Good ESG performance of listed companies significantly enhances investor attention, and investor attention plays an intermediary role in reducing abnormal stock price volatility.

## 4 Research design

### 4.1 Sample selection and data sources

This paper selects A-share listed companies from 2011 to 2020 as its research object, and carries out the following screening in accordance with the practice of existing literature research: (1) Companies in special industries such as finance and insurance, are excluded due to their unique business models and risk characteristics, which may affect the generality of the research results if included; (2) Companies with missing key variable data are excluded to avoid significant biases in the research results; (3) ST, *ST and other operating abnormal companies are excluded as the financial data and stock price volatility of such companies are significantly different from those of normal operations. Meanwhile, to prevent the influence of outliers in the data, continuous variables are trimmed to the 1st and 99th percentiles. Data sources include ESG ratings and investor attention data from the Wind database, while the rest are from the Cathay Pacific CSMAR database. Statistical analysis of the data is conducted using Stata17 software, which is commonly used in empirical economics research.

### 4.2 Definition of variables

**4.2.1 Explained variables.** The explanatory variable in this study is stock price volatility (Vol). According to existing research [9], the mean monthly stock return variance is used to measure the stock price volatility of listed companies. This is calculated by first adjusting for the effect of overall market returns in the current period, followed by the calculation of the monthly stock return variance based on the variance of daily individual stock returns after adjustment. Finally, the average monthly stock return variance across the 12 months of the year is calculated. To minimize insensitivity to the too-small numbers, the result is multiplied by 100 to serve as a measure of each company's stock price volatility.

**4.2.2 Explanatory variables.** The explanatory variable is corporate ESG performance (ESG). Most of the current literature uses ESG ratings to measure corporate ESG performance. In China, prominent ESG evaluation institutions include Huazheng, Shangdao Ronglv, and Bloomberg, although the latter two disclose comparatively less data. Referring to existing studies [1], this paper adopts the Huazheng ESG ratings as a proxy for corporate ESG performance. Huatai Securities Company has assessed the ESG performance of A-share and debt issuers since 2009, and now it has covered all A-share listed companies with the most comprehensive data. The Huazheng index is widely recognized in both industry and academia [39]. Huazheng ratings range from AAA to C across 9 levels and are updated quarterly. In this paper, these ratings are scored from 1 to 9, and the arithmetic average of the 4 quarterly ratings indicates the annual ESG performance of firms. In the robustness test, Bloomberg ESG ratings are substituted for the explanatory variables.

**4.2.3 Mediating variables.** The mediating variable is Investor Attention (Att). With the rapid development of the Internet, an increasing number of investors use search engines to monitor company stocks. Some scholars first used Google search volume to measure investor attention in 2011, where higher search data indicated greater investor attention to stocks [40]. Since then, scholars have made increasing use of the Internet search index to measure investor

**Table 1. Variable description and definition.**

| Variable Type | Variable Name | Variable Symbol | Variable Definition |
|---|---|---|---|
| Explained Variable | Stock price fluctuation | Vol | Mean of variance of firm's monthly stock price returns |
| Explanatory variable | Corporate ESG performance | ESG | The ESG ratings of CSI are assigned from 1 to 9 on a scale of 1 to 9, averaged over 4 quarters of the year. |
| Intermediary variable | Investor attention | Att | Baidu search index, plus 1 to take the natural logarithm |
| Control variable | Enterprise size | Size | The total assets of the company are taken as a natural logarithm |
| | Gearing | Lev | Total liabilities/total assets |
| | Return on net assets | Roa | Net profit/total assets |
| | Growth capacity | Growth | (Current year's operating income—previous year's operating income)/previous year's operating income |
| | Free cash flow | Cf | Net cash flows from operating activities/total assets |
| | Age of business | Age | Natural logarithm of the time of incorporation |
| | Book-to-market ratio | BM | Carrying value divided by market value |
| | Proportion of independent directors | Ind | Ratio of independent directors to the number of board members |
| | Integration of two positions | Dual | If the chairman of the board of directors and general manager is the same person then take the value of 1, otherwise take the value of 0 |
| | Proportion of managerial holdings | Mhold | Number of shares held by directors and supervisors/total shares |
| | Equity balance | Balance | Shareholding of 2nd-5th largest shareholder / Shareholding of 1st largest shareholder |
| | Particular year | Year | Year dummy variable |
| | Industry | Industry | Industry dummy variables |

attention. Foreign scholars commonly use Google search volume data to construct indicators of investor attention, while Chinese scholars commonly use the Baidu search index. Recent studies that have used fixed-effects constant coefficient panel data models empirically demonstrate that the Baidu index, specifically when searching for security codes, is an effective indicator of investor attention, while the Baidu index for searching security abbreviations tends to be noisier, which may be the result of shareholders' attention to non-investment-related company information [41]. Referring to this methodology, this paper uses the stock codes of Chinese listed companies as search keywords to generate the Baidu index, with the natural logarithmic of the index value, adjusted by adding 1, as the mediating variable, denoted as Investor Attention (Att).

**4.2.4 Control variables.** Based on existing studies [1], this paper selects several control variables that may affect the firm share price volatility. These variables include firm size (Size), gearing ratio (Lev), return on assets (Roa), growth capacity (Growth), free cash flow (Cf), firm age (Age), book-to-market ratio (BM), proportion of independent directors (Ind), dual-class share structure (Dual), proportion of managerial shareholding (Mhold), and checks and balances on shareholding (Balance). To account for temporal and sectoral variations, the paper controls for both year (Year) and industry (Industry) fixed effects. Detailed definitions and measures of each variable are provided in Table 1.

### 4.3 Model setting

To test the effect of firms' ESG performance on stock price volatility (H1), the following model is constructed.

$$\text{Vol}_{i,t} = \beta_0 + \beta_1 \text{ESG}_{i,t} + \beta_2 \sum \text{Controls}_{i,t} + \sum \text{Year} + \sum \text{Industry} + \varepsilon_{i,t} \qquad (1)$$

To test the effect of corporate ESG performance on stock price volatility through investor attention (H2), the following model is constructed.

$$\text{Att}_{i,t} = \beta_0 + \beta_1 \text{ESG}_{i,t} + \beta_2 \sum \text{Controls}_{i,t} + \sum \text{Year} + \sum \text{Industry} + \varepsilon_{i,t} \qquad (2)$$

$$\text{Vol}_{i,t} = \beta_0 + \beta_1 \text{ESG}_{i,t} + \beta_2 \text{Att}_{i,t} + \beta_3 \sum \text{Controls}_{i,t} + \sum \text{Year} + \sum \text{Industry} + \varepsilon_{i,t} \qquad (3)$$

Where: i represents the company, t represents the year, and Vol represents the explanatory variable stock price volatility. ESG represents the explanatory variable corporate ESG performance as measured by Huazheng ESG ratings, Att represents the mediator variable investor concern, ∑Control represents a set of control variables that may affect the company's stock price volatility, and ∑Year and ∑Industry denote the fixed annual and Industry, and ε represents the random error term.

## 5 Empirical analysis

### 5.1 Descriptive statistical results

The descriptive statistical analysis in Table 2 shows that stock price volatility (Vol) ranges from a minimum of 0.214 to a maximum of 12.59, with a standard deviation of 1.411, indicating a substantial difference in stock price volatility among listed companies. The mean value of a core explanatory variable (ESG) is 6.469, with a standard deviation of 1.133, indicating that listed companies generally receive ratings between BBB and A, which is a medium to high level of ESG performance. There is a significant variation in ESG ratings among companies. Investor attention (Att) has a mean value of 0.403, from a minimum of 0 to a maximum of 32.40, indicating a substantial difference in investor attention across different listed companies.

### 5.2 Correlation analysis

Table 3 shows the results of correlation analysis for each variable. As shown in Panel A, corporate ESG performance (ESG) exhibits a significant negative correlation with stock price volatility (Vol) at the 1% level, indicating that good corporate ESG performance can significantly inhibit abnormal stock price volatility, preliminarily verifying Hypothesis 1. Additionally, stock price volatility (Vol) is significantly correlated with all control variables at the 1% level,

**Table 2. Descriptive statistics.**

| Variable | Obs | Mean | Standard Deviation | Minimum | Upper Quartile | Maximum |
|---|---|---|---|---|---|---|
| Vol | 29266 | 1.436 | 1.411 | 0.214 | 1.105 | 12.59 |
| ESG | 29266 | 6.469 | 1.133 | 1 | 6 | 9 |
| Att | 29266 | 0.403 | 0.629 | 0 | 0.276 | 32.40 |
| Size | 29266 | 22.75 | 1.168 | 19.89 | 22.58 | 28.73 |
| Lev | 29266 | 0.418 | 0.210 | 0.0509 | 0.408 | 0.929 |
| Roa | 29266 | 0.0368 | 0.0666 | -0.299 | 0.0381 | 0.199 |
| Cf | 29266 | 0.0458 | 0.0695 | -0.168 | 0.0455 | 0.240 |
| Growth | 29266 | 0.184 | 0.342 | -0.336 | 0.0955 | 2.021 |
| Age | 29266 | 10.03 | 7.641 | 0 | 9 | 30 |
| Ind | 29266 | 0.376 | 0.053 | 0.333 | 0.364 | 0.571 |
| Dual | 29266 | 0.697 | 0.460 | 0 | 1 | 1 |
| Mhold | 29266 | 9.032 | 14.49 | 0 | 0.020 | 80.01 |
| Balance | 29266 | 0.367 | 0.286 | 0.0105 | 0.288 | 0.994 |

**Table 3. Correlation analysis results.**

| | | | | | | Panel A | | | | | | |
|---|---|---|---|---|---|---|---|---|---|---|---|---|
| | **Vol** | **ESG** | **Size** | **Lev** | **Roa** | **Cf** | **Growth** | **Age** | **Ind** | **Dual** | **Mhold** | **Balance** |
| Vol | 1 | | | | | | | | | | | |
| ESG | -0.118*** | 1 | | | | | | | | | | |
| Size | -0.146*** | 0.355*** | 1 | | | | | | | | | |
| Lev | -0.093*** | 0.069*** | 0.420*** | 1 | | | | | | | | |
| Roa | -0.005*** | 0.159*** | 0.066*** | -0.369*** | 1 | | | | | | | |
| Cf | -0.031*** | 0.094*** | 0.145*** | -0.160*** | 0.375*** | 1 | | | | | | |
| Growth | 0.252*** | -0.003 | -0.058*** | -0.137*** | 0.296*** | -0.023*** | 1 | | | | | |
| Age | -0.200*** | 0.177*** | 0.404*** | 0.377*** | -0.184*** | -0.017*** | -0.285*** | 1 | | | | |
| Ind | 0.019*** | -0.012** | 0.012** | -0.011* | -0.017*** | -0.006 | 0.006 | -0.027*** | 1 | | | |
| Dual | -0.106*** | 0.095*** | 0.143*** | 0.130*** | -0.030*** | 0.018*** | -0.110*** | 0.227*** | -0.121*** | 1 | | |
| Mhold | 0.135*** | -0.124*** | -0.301*** | -0.278*** | 0.136*** | 0.003 | 0.199*** | -0.485*** | 0.102*** | -0.271*** | 1 | |
| Balance | 0.067*** | -0.063*** | -0.056*** | -0.090*** | -0.010* | -0.011* | 0.071*** | -0.113*** | -0.014** | -0.042*** | 0.047** | 1 |
| | | | | | | Panel B | | | | | | |
| VIF test | **ESG** | **Size** | **Lev** | **Roa** | **Cf** | **Growth** | **Age** | **Ind** | **Dual** | **Mhold** | **Balance** | **Mean** |
| | 1.31 | 1.97 | 1.85 | 1.62 | 1.32 | 1.24 | 1.80 | 1.05 | 1.14 | 1.46 | 1.06 | 1.44 |

Note

***, ** and * indicate significance at the 1%, 5% and 10% levels.

with correlation coefficients below 0.5. Meanwhile, the variance inflation factor (VIF) for all variables is less than 5, as shown in Panel B in Table 3, indicating no serious multicollinearity problem, confirming the robustness of variable selection in the model.

## 5.3 Baseline regression analysis

In Table 4, columns (1)-(4) present the basic regression results of the impact of firms' ESG performance on stock price volatility. In column (1), where no control variables are included and year and industry fixed effects are not controlled, the regression coefficient for ESG is -0.1474, significant at the 1% level. In column (2), without control variables but with year and industry fixed effects controlled, the regression coefficient for ESG is -0.1299, also significant at the 1% level. Columns (3) and (4) introduce control variables; column (3) does not control for fixed effects, and column (4) controls for year and industry fixed effects. In column (3), the regression coefficient for ESG is -0.0762, significant at the 1% level, while in column (4), it is -0.0450, also significant at the 1% level. The results under four different scenarios all indicate that corporate ESG performance has a significant negative impact on stock price volatility, i.e., the better the corporate ESG performance, the smaller the abnormal stock price volatility among listed companies. Thus, the regression results verify hypothesis H1.

## 5.4 Analysis of intermediation effects

To examine the mediating role of investor attention between corporate ESG performance and stock price volatility, this paper uses stepwise regression [42]. The results are shown in columns (4)-(6) of Table 4. Column (4) shows that the regression coefficient of corporate ESG performance on stock price volatility (Vol) is -0.0450 (i.e., the total effect), significant at the 1% level, indicating a significantly negative relationship. Column (5) shows the regression results of corporate ESG performance on investor attention, with a significant positive coefficient of 0.0125 at the 1% level, indicating that the better the corporate ESG performance, the higher

**Table 4. Baseline regression test results.**

| VARIABLES | (1) Vol | (2) Vol | (3) Vol | (4) Vol | (5) Att | (6) Vol |
|---|---|---|---|---|---|---|
| ESG | -0.1474*** | -0.1299*** | -0.0762*** | -0.0450*** | 0.0125*** | -0.0438*** |
|  | (-27.7594) | (-24.8345) | (-12.5663) | (-7.5054) | (4.8345) | (-7.2908) |
| Att |  |  |  |  |  | -0.0960*** |
|  |  |  |  |  |  | (3.4993) |
| Size |  |  | -0.0605*** | -0.1814*** | 0.2114*** | -0.1980*** |
|  |  |  | (-7.2329) | (-19.6583) | (28.8141) | (-19.5002) |
| Lev |  |  | -0.1436*** | 0.2299*** | -0.0635*** | 0.2349*** |
|  |  |  | (-3.0494) | (4.8373) | (-2.6810) | (4.9455) |
| Roa |  |  | -2.0690*** | -1.1092*** | -0.3366*** | -1.0828*** |
|  |  |  | (-13.1568) | (-7.3188) | (-6.5971) | (-7.1467) |
| Cf |  |  | 0.4116*** | 0.1267 | 0.0610 | 0.1219 |
|  |  |  | (3.0507) | (0.9649) | (1.2188) | (0.9288) |
| Growth |  |  | 0.9895*** | 0.9327*** | -0.0505*** | 0.9367*** |
|  |  |  | (19.7019) | (19.9345) | (-5.9259) | (19.9985) |
| Age |  |  | -0.0166*** | -0.0233*** | 0.0017** | -0.0235*** |
|  |  |  | (-13.3253) | (-17.8398) | (2.5263) | (-17.9268) |
| Ind |  |  | 0.1949 | -0.0909 | 0.5009*** | -0.1303 |
|  |  |  | (1.4180) | (-0.7038) | (6.0212) | (-1.0043) |
| Dual |  |  | -0.1239*** | -0.0565*** | -0.0253*** | -0.0546*** |
|  |  |  | (-6.2499) | (-3.0511) | (-3.5614) | (-2.9462) |
| Mhold |  |  | 0.0015** | -0.0017** | -0.0001 | -0.0017** |
|  |  |  | (1.9893) | (-2.3637) | (-0.4141) | (-2.3572) |
| Balance |  |  | 0.1384*** | 0.0399 | -0.0034 | 0.0402 |
|  |  |  | (5.1555) | (1.5810) | (-0.3329) | (1.5927) |
| _cons | 2.3896*** | 2.1688*** | 3.3554*** | 5.2731*** | -4.5046*** | 5.6271*** |
|  | (64.1640) | (16.7805) | (19.3035) | (24.0954) | (-25.1848) | (23.7259) |
| Year | No | Yes | No | Yes | Yes | Yes |
| Industry | No | Yes | No | Yes | Yes | Yes |
| N | 29266 | 29266 | 29266 | 29266 | 29266 | 29266 |
| $R^2$ | 0.0140 | 0.1532 | 0.1033 | 0.2422 | 0.2449 | 0.2431 |

Note: Heteroscedasticity robust standard errors in parentheses

***, ** and * indicate significance at the level of 1%, 5% and 10%

the investor attention. In column (6), the regression coefficient for investor attention (Att), which is -0.0960, indicates the effect of investor attention on stock price volatility after controlling for the effect of ESG. Meanwhile, the regression coefficient for ESG, which is -0.0438, indicates the effect of ESG on stock price volatility (i.e., the direct effect) after controlling for the mediating variable, Att. The indirect effect is calculated as -0.0960 × 0.0125 = -0.0012. The coefficients of both the direct and indirect effects are significant at the 1% level with the same negative sign, indicating that investor concern plays a partially intermediary role between firms' ESG performance and stock price volatility, thus verifying hypothesis H2.

## 5.5 Robustness tests

**5.5.1 Replacement of explanatory variable measures.** ESG ratings currently lack a standardized measure, leading to differences in evaluation standards and coverage among different

rating agencies. To mitigate the possible impact of ESG ratings data selection on benchmark regression results, this paper replaces explanatory variables with Bloomberg ESG ratings. These ratings are measured by dividing the ESG scores of listed companies in mainland China, as released by Bloomberg, by 100, denoted by ESG_b. Referring to the existing literature [43], simplified test results of the mediation effect are shown in columns (1) and (2) of Table 5. The regression coefficients remain unchanged in terms of sign and significance, which demonstrates the robustness of the previous conclusions.

**Table 5. Robustness test results.**

| VARIABLES | (1) Vol | (2) Att | (3) V0l_ram | (4) Att_ia | (5) Vol |
|---|---|---|---|---|---|
| ESG_b | -0.0035*** | 0.0053*** | | | |
| | (-3.3702) | (3.5994) | | | |
| L. Vol | | | | | -0.1646*** |
| | | | | | (-20.0859) |
| L2.Vol | | | | | -0.3143*** |
| | | | | | (-39.4765) |
| L3.Vol | | | | | -0.0786*** |
| | | | | | (-12.5138) |
| ESG | | | -0.0831*** | 0.0006*** | -0.2790*** |
| | | | (-8.9447) | (3.2910) | (-18.6447) |
| Size | -0.0975*** | 0.2796*** | -0.0242 | -0.0115*** | 1.0219*** |
| | (-10.0952) | (19.3362) | (-1.4009) | (-39.8816) | (46.5472) |
| Lev | 0.3023*** | -0.0861* | -0.2440*** | 0.0103*** | 0.0400 |
| | (4.9105) | (-1.6893) | (-3.3620) | (6.9929) | (0.4253) |
| Roa | -0.8700*** | -0.6602*** | -1.1742*** | 0.0190*** | -1.1566*** |
| | (-3.9996) | (-4.1839) | (-7.6728) | (4.8226) | (-8.8797) |
| Cf | 0.2769* | 0.0909 | 0.5277*** | -0.0071 | -0.1870 |
| | (1.7117) | (0.8841) | (4.3392) | (-1.6348) | (-1.5222) |
| Growth | 0.2330*** | -0.1156*** | 0.6544*** | 0.0303*** | -0.1359*** |
| | (5.0752) | (-3.6426) | (15.9857) | (24.4692) | (-4.7751) |
| Age | -0.0047*** | -0.0023 | -0.0849*** | -0.0014*** | -0.0384*** |
| | (-2.7456) | (-1.2217) | (-21.4815) | (-30.4340) | (-11.8442) |
| Ind | -0.0672 | 1.1112*** | -0.0765 | -0.0061 | 1.0143*** |
| | (-0.5441) | (5.5569) | (-0.4483) | (-1.3604) | (4.3702) |
| Dual | -0.0124 | -0.0019 | -0.0426* | -0.0014** | 0.0067 |
| | (-0.6247) | (-0.1162) | (-1.8328) | (-2.2996) | (0.2604) |
| Mhold | 0.0004 | 0.0006 | -0.0000 | 0.0001*** | 0.0022 |
| | (0.4737) | (0.8438) | (-0.0078) | (3.6675) | (1.2094) |
| Balance | 0.0638** | 0.0488** | 0.0776** | 0.0002 | -0.0116 |
| | (2.3971) | (2.0189) | (2.0683) | (0.2562) | (-0.1864) |
| _cons | 3.0363*** | -6.3224*** | 4.5599*** | 0.2913*** | -19.6717*** |
| | (12.2854) | (-17.2055) | (8.9758) | (40.6430) | (-42.0757) |
| Year | Yes | Yes | Yes | Yes | Yes |
| Industry | Yes | Yes | Yes | Yes | Yes |
| N | 9075 | 9075 | 29266 | 29266 | 17461 |
| $R^2$ | 0.3107 | 0.2972 | 0.2879 | 0.2674 | 4118.15 (Wald chi2) |

Note

***, ** and * indicate significance at the 1%, 5% and 10% levels.

**5.5.2 Substitution of explanatory variables.** Referring to a previous study [9], the measure of share price volatility for the explanatory variable is replaced with the calculation of the annual individual stock return variance based on the original return of individual stocks, denoted as V0l_ram. The regression results are shown in column (3) of Table 5, where the regression coefficient is -0.0831, significant at the 1% level, reflecting a significantly negative correlation between the ESG performance of listed companies and share price volatility, which indicates the robustness and stability of the conclusions in this study.

**5.5.3 Substitution of mediator variables.** Referring to the existing literature [44], the average turnover rate over the 30 trading days prior to the earnings announcement is used as a proxy variable (Att_ia) for the mediator variable Att. The regression results are shown in column (4) of Table 5, with a regression coefficient of 0.0006, which is significantly positive at the 1% level. This substitution further validates the robustness of the conclusion in this study.

**5.5.4 GMM methods.** To verify the robustness of the dynamic panel data model, this study uses the Generalized Method of Moments (GMM) for testing. In the model, L.Vol indicates the lagged explanatory variable of stock price volatility by one period, L2.Vol by two periods, and L3. Vol by three periods, which are included in the regression model as the significant explanatory variables of the current stock price volatility. The results are shown in Column (5) of Table 5, where the regression coefficient is -0.2790, significantly negative at the 1% level, indicating that corporate ESG performance has a significant negative impact on stock price volatility. The conclusion of the study still holds, which further validates the robustness and reliability of the model.

## 5.6 Endogeneity tests and corrections

**5.6.1 Explanatory variables lagged by one period.** To explore the potential causality between firms' ESG performance and stock volatility, and to address the resulting endogeneity problem, this study retests the main hypotheses H1 and H2 by lagging the explanatory variables by one period (denoted as LESG). The regression results are shown in columns (1) and (2) of Table 6, with regression coefficients of -0.0676 and 0.0119. Both coefficients are significant at the 1% level, indicating that the previous conclusion still holds after considering the endogeneity problem triggered by the inversion of causality, i.e., the better the ESG performance, the higher the investor attention, and the lower the stock price volatility.

**5.6.2 Propensity score matching method (PSM).** Firms' ESG performance may be affected by their internal governance and financial status, i.e., there are potential self-selection biases in the sample. To mitigate this endogeneity problem, this study adopts the propensity score matching method (PSM). Firstly, sample data is divided into two groups, i.e., the group with higher and lower corporate ESG performance, based on the mean value of the explanatory variable corporate ESG performance; and then, with the control variable as the matching variable, the sample firms are matched in a 1:1 ratio using the nearest neighbor matching method. Columns (3) and (4) of Table 6 show the regression results after using PSM. The empirical results are consistent with the results from the baseline regression analysis.

**5.6.3 Instrumental variables approach.** To address the endogeneity issue in the regression model, this study uses the instrumental variable method. In reference to existing research [45], the mean ESG performance of the company's industry (Mean1) and the mean ESG performance of the company's city (Mean2) are used as instrumental variables, which satisfy the relevance requirement of instrumental variables because companies in the same industry or the same city tend to face similar external environments and governmental policies. The mean ESG performance of companies in the same industry or city is usually not directly related to the company's stock price volatility, and the above two variables fulfill the exogeneity requirement of instrumental variables. The test results based on the instrumental variable method are

**Table 6. Endogeneity test results.**

| VARIABLES | (1) Vol | (2) Att | (3) Vol | (4) Att | (5) Vol | (6) Att |
|---|---|---|---|---|---|---|
| L.ESG | -0.0676*** | 0.0119*** | | | | |
| | (-14.9001) | (4.0514) | | | | |
| ESG | | | -0.0633*** | 0.0070* | -0.0640*** | 0.0178** |
| | | | (-8.0686) | (1.8165) | (-3.1574) | (1.9716) |
| Size | -0.0620*** | 0.2193*** | -0.1869*** | 0.1694*** | -0.1744*** | 0.2094*** |
| | (-12.8802) | (26.4720) | (-13.6088) | (22.2873) | (-15.7010) | (42.3439) |
| Lev | 0.1871*** | -0.0524** | 0.3052*** | -0.0628** | 0.2187*** | -0.0604*** |
| | (6.3465) | (-2.0296) | (4.4544) | (-1.9823) | (4.5605) | (-2.8273) |
| Roa | -1.2987*** | -0.2843*** | -0.8243*** | -0.2270*** | -1.0758*** | -0.3460*** |
| | (-12.8446) | (-4.9388) | (-3.8937) | (-2.9008) | (-7.6220) | (-5.5052) |
| Cf | 0.2327*** | 0.0188 | 0.1961 | -0.0183 | 0.1299 | 0.0601 |
| | (2.8803) | (0.3145) | (1.1102) | (-0.2467) | (1.0952) | (1.1379) |
| Growth | 0.1798*** | -0.1133*** | 0.9294*** | -0.0482*** | 0.9325*** | -0.0504*** |
| | (9.0103) | (-8.4562) | (14.1012) | (-3.7573) | (39.9406) | (-4.8495) |
| Age | -0.0053*** | 0.0023*** | -0.0221*** | 0.0024*** | -0.0232*** | 0.0017*** |
| | (-7.3069) | (3.0722) | (-11.8928) | (3.3474) | (-18.3067) | (2.9838) |
| Ind | 0.2220*** | 0.5563*** | 0.1037 | 0.2372*** | -0.0932 | 0.5015*** |
| | (2.7565) | (5.8899) | (0.5810) | (2.6360) | (-0.6702) | (8.0951) |
| Dual | -0.0204** | -0.0237*** | -0.0372 | -0.0153* | -0.0551*** | -0.0257*** |
| | (-2.0571) | (-2.9596) | (-1.4614) | (-1.7406) | (-3.2893) | (-3.4438) |
| Mhold | 0.0009** | 0.0001 | -0.0021** | -0.0002 | -0.0017*** | -0.0001 |
| | (2.3992) | (0.2959) | (-1.9807) | (-0.6869) | (-2.8553) | (-0.2932) |
| Balance | 0.0405*** | 0.0006 | 0.0584* | -0.0301** | 0.0378 | -0.0028 |
| | (2.7622) | (0.0502) | (1.6783) | (-2.1939) | (1.4560) | (-0.2413) |
| _cons | 3.8402*** | -4.3968*** | 5.4787*** | -3.4675*** | 5.2485*** | -4.4977*** |
| | (34.2048) | (-22.5623) | (15.9742) | (-19.0660) | (26.8542) | (-51.6845) |
| Year | Yes | Yes | Yes | Yes | Yes | Yes |
| Industry | Yes | Yes | Yes | Yes | Yes | Yes |
| Phase I F-value | | | | | 150.59 | |
| N | 24763 | 24763 | 13780 | 13780 | 29266 | 29266 |
| R² | 0.3164 | 0.2474 | 0.2504 | 0.2100 | 0.2420 | 0.2448 |

Note

***, ** and * indicate significance at the 1%, 5% and 10% levels.

shown in columns (5) and (6) of Table 6. The first-stage F-value for the weak instrumental test is 150.59, which is much higher than 10, indicating no weak instrumental variable problem. Moreover, the core explanatory variables are highly correlated with the instrumental variables. In the second stage regression, the results show that ESG performance is significantly negatively correlated with stock price volatility and that ESG is significantly positively correlated with investor concern. The conclusion remains consistent with the previous section.

## 6. Heterogeneity analysis

### 6.1 Heterogeneity analysis of enterprise life cycles

According to the life cycle theory, an enterprise undergoes four stages: start-up, growth, maturity and decline. Companies in different life cycles have significant differences in ESG

**Table 7. Heterogeneity analysis results.**

| VARIABLES | (1) Growth stage Vol | (2) Maturity stage Vol | (3) Bull market Vol | (4) Bear market Vol |
|---|---|---|---|---|
| ESG | -0.0810*** | 0.0114 | -0.0117 | -0.0647*** |
|  | (-12.0539) | (0.7708) | (-1.5758) | (-8.2132) |
| Size | -0.0723*** | -0.2643*** | -0.2455*** | -0.1532*** |
|  | (-8.8995) | (-15.5965) | (-21.8849) | (-12.8949) |
| Lev | 0.2677*** | 0.0500 | 0.4068*** | 0.1480** |
|  | (5.8321) | (0.5093) | (6.6936) | (2.3457) |
| Roa | -1.2240*** | 0.0964 | 0.1339 | -1.6892*** |
|  | (-8.8678) | (0.3133) | (0.5806) | (-8.8736) |
| Cf | 0.4967*** | -0.1848 | 0.0357 | 0.0750 |
|  | (3.8729) | (-0.7435) | (0.2478) | (0.4149) |
| Growth | 0.3827*** | 0.8025*** | 0.3776*** | 1.2832*** |
|  | (8.4170) | (20.9440) | (12.6648) | (17.8875) |
| Age | -0.0062*** | -0.0435*** | -0.0144*** | -0.0251*** |
|  | (-5.2027) | (-17.2184) | (-9.3464) | (-14.5186) |
| Ind | 0.3396*** | -0.6378** | 0.1415 | -0.1798 |
|  | (2.5886) | (-2.3632) | (0.9248) | (-1.0312) |
| Dual | -0.0261 | -0.0734** | -0.0212 | -0.0673*** |
|  | (-1.6149) | (-2.3365) | (-1.1126) | (-2.6383) |
| Mhold | 0.0012** | -0.0036*** | -0.0012 | -0.0017* |
|  | (2.0596) | (-3.3370) | (-1.4522) | (-1.6951) |
| Balance | 0.0339 | 0.0627 | -0.0085 | 0.0639* |
|  | (1.3946) | (1.2424) | (-0.3046) | (1.8510) |
| _cons | 2.8540*** | 7.0860*** | 8.1886*** | 4.6710*** |
|  | (15.5784) | (18.8366) | (30.4848) | (16.4690) |
| Year | Yes | Yes | Yes | Yes |
| Industry | Yes | Yes | Yes | Yes |
| N | 11616 | 13133 | 9552 | 19714 |
| $R^2$ | 0.2733 | 0.2741 | 0.3886 | 0.2331 |

Note

***, ** and * indicate significance at the 1%, 5% and 10% levels.

performance, which may have varying impacts on stock price volatility. Considering the listing requirements for A-share Chinese companies, those listed have generally moved beyond the start-up stage; research on companies in the decline stage, which typically have poor ESG performance, may not yield meaningful insights. Referring to existing research [46], this study screens the sample firms in the growth and maturity stages. Regression results in columns (1) and (2) of Table 7 show that the regression coefficient of ESG in the growth period is -0.0810, significantly negative at the 1% level, while the coefficient of ESG in the maturity period is 0.0114, indicating no significant impact on stock price volatility. For listed companies in the growth period, there is a significant negative correlation between better ESG performance and reduced abnormal stock price volatility, but for listed companies in the maturity period, the relationship between ESG performance and stock price volatility is not significant. The reason is companies in the growth stage generally tend to invest their limited funds in new market development [47]. If a company can perform well in ESG despite a lack of resources, it

indicates a strong sustainable development ability and determination, which obviously exceeds investor expectations. Thus, such companies are more likely to attract long-term investor attention and shareholding, significantly reducing abnormal stock price volatility. In contrast, companies in the maturity stage have abundant financial resources, and good ESG performance is in line with investors' psychological expectations, leading to an insignificant correlation with stock price volatility.

### 6.2 Heterogeneity analysis of bull and bear markets

Numerous studies have shown that investor attention to listed companies and the volatility of their stock prices vary significantly between bull and bear markets. Referring to existing research [7], this paper classifies the years 2014–2017 as a bull market and other years are classified as bear markets to control for systematic risk impacts on the study's outcomes. The regression results are shown in columns (3) and (4) of Table 7. The coefficient for ESG in the bull market state (column 3) is -0.0117, which is not significant, while the coefficient for ESG in the bear market state (column 4) is -0.0647, which is significant at the 1% level. This indicates that the better the ESG performance of listed companies in the bear market, the smaller the abnormal stock price volatility, indicating a significant negative correlation between the two variables. The reason is that in the bear market, investor confidence tends to be low. Companies with good ESG performance send positive signals to the outside world that they are capable of sustainable development, so investors are more willing to hold onto these stocks for longer periods, thus reducing irrational investment behavior and abnormal stock price volatility. In bull markets, stock prices of listed companies tend to rise, consequently expanding investment options. Short-term profit-seeking behavior among retail investors increases, which reduces the perceived advantages of sustainable development inherent to companies with strong ESG performance, thus leading to an insignificant impact of ESG performance on stock price volatility in the bull market.

## 7. Conclusions and recommendations

### 7.1 Conclusions and discussion

This study's empirical analysis of Chinese A-share listed companies from 2011 to 2020 revealed that good ESG performance significantly mitigates abnormal stock price volatility. This finding is consistent with the mainstream views in existing literature. Some studies suggest that ESG performance effectively suppresses stock price volatility, with innovation efficiency as a key mediating mechanism [2]. Enterprise ESG information disclosure can reduce stock price volatility by increasing information transparency, internal control quality, institutional shareholding ratios [19] or analyst attention [20]. Another study found a significant negative correlation between ESG ratings of industrial enterprises and stock price volatility, which was not observed in non-industrial enterprises [48]. Conversely, studies with ESG scores as proxies for the social responsibility of American banks have identified a significant positive correlation between ESG scores and stock price volatility, attributed to the increased financial burden of long-term social responsibility investments by banks [3]. Other studies argue that ESG performance does not markedly influence abnormal stock price volatility [21], especially in developing countries like Indonesia, where investor protection and regulatory enforcement are weak, and the inhibitory effect of ESG on stock price fluctuations is not significant [4]. Regarding the impact of the Environmental (E), Social (S), and Governance (G) dimensions on stock price volatility, some studies indicate that the Environmental dimension exerts a negligible influence on stock price volatility, whereas the Social and Governance dimensions significantly mitigate abnormal stock price volatility [14]. Conversely, research posits that

Environmental performance positively correlates with stock price volatility, Governance practices inversely relate to stock price volatility, and Social performance does not markedly affect stock price volatility [49]. It is evident that there are significant differences in the research results of existing literature regarding whether ESG performance affects abnormal stock price fluctuations.

This study uniquely explores the mechanism by which corporate ESG performance influences stock price volatility, with investor attention as a mediating factor. The broad adoption of ESG practices in listed companies has markedly increased investor attention. Concerning the effect of investor attention on stock price volatility, academic opinions are polarized between the "aggravation theory" [23] and the "stabilization theory" [24]. This study investigates whether enhanced ESG performance in listed companies, by attracting more investor attention, serves to exacerbate or mitigate stock price volatility, an area that remains underexplored in current research. Our findings reveal that investor attention partially mediates the effect of corporate ESG performance in reducing abnormal stock price volatility. Notably, during growth phases and in bear markets, robust ESG performance significantly curtails abnormal stock price movements, a phenomenon less evident in maturity phases and bull markets. This study fills existing research gaps by shedding light on the role of investor attention in the relationship between ESG performance and stock price volatility, thereby enriching both empirical and theoretical understandings of this dynamic. The implications of these findings are significant for listed companies, investors, and regulatory bodies, underscoring the role of ESG practices in sustainable corporate growth and capital market stability.

## 7.2 Recommendations

For listed companies, ESG investment and rating should be emphasized to curb abnormal stock price volatility by enhancing corporate ESG performance. Specific practices are as follows: (1) ESG investment in conjunction with enterprise life cycles, especially in the growth period, should focus on increasing environmental investment, promoting green transformation, actively undertaking social responsibility, improving internal governance, and enhancing ESG ratings; (2) it is crucial to strengthen ESG information disclosure through a variety of network platforms so that external investors can fully understand the ESG performance of the company through network search, thus reducing information asymmetry; (3) In the bear market, greater attention should be paid to good corporate ESG performance to send a signal that the company has a strong sustainable development ability. This can attract more investor attention, thus significantly suppressing abnormal stock price volatility, returning the company's stock price to its intrinsic value, and facilitating steady value growth and high-quality development.

For investors, (1) when choosing investment target companies, in addition to the financial indicators, they should also pay close attention to the ESG ratings of the companies through a variety of online platforms. It is crucial to fully interpret the signals conveyed by the disclosure of ESG information and actively invest in listed companies with good ESG performance. This approach can reduce irrational investment behaviors and minimize losses caused by abnormal share price volatility. (2) The concept of long-term value investment should be adhered to regardless of market conditions. Especially in bear markets, more investments should be made in companies with good ESG performance. (3) It is necessary to pay attention to the life cycle of listed companies and focus on investments in listed companies in the growth period with good ESG performance to continuously improve investment returns.

For the government, (1) government subsidies, tax incentives and other supportive policies can be utilized to encourage listed companies to undertake environmental protection and

social responsibility. Targeted efforts should be made to improve the existing mandatory ESG information disclosure system, sustainable development capabilities, and high-quality economic development. (2) It is suggested to limit pollution emissions through a sewage permit management system and increase penalties for companies "greenwashing" behavior. It is equally important to compel listed companies to focus on ESG investments and incentivize companies' environmental governance through market means such as sewage charges or sewage rights trading to improve their ESG performance. (3) The ESG ratings of listed companies should be regularly disclosed on the official platform free of charge. This will facilitate investors' access to accurate ESG information, avoid irrational investment operations due to asymmetric information, reduce abnormal stock price volatility, improve investor protection regulations, and promote the healthy development of the capital market.

### 7.3 Research limitations and future research prospects

This paper empirically analyzes data from Chinese A-share listed companies from 2011 to 2020 and finds that good corporate ESG performance significantly reduces abnormal stock price volatility, with investor attention as a partly positive mediating factor. This study fills a gap in the existing literature by considering the mediating effect of investor attention, but there are some limitations. On the one hand, the current ESG rating lacks an authoritative and unified evaluation system. Common ESG evaluation systems include Huazheng, Shangdao Ronglv, Bloomberg, Wind, etc. Differences in the selected ESG rating systems may lead to varying results. On the other hand, this paper uses the Baidu index of stock codes to measure investor concern, which is more effective than using the Baidu index of stock names, but it is still difficult to accurately measure investor attention.

Looking ahead, future research should construct a more authoritative ESG rating framework and seek more accurate metrics to quantify investor concerns. This would enable a more effective exploration of the mechanisms by which ESG performance affects abnormal stock price volatility. In addition, a detailed study of ESG performance across different industries and regions, as well as an in-depth analysis of the impact of each ESG subcomponent on abnormal stock price volatility, could further reveal the relationship between ESG performance and stock price volatility. These efforts will not only enrich the theoretical research on the economic consequences of ESG performance but also provide empirical evidence on how stakeholders can utilize ESG performance in response to abnormal stock price volatility.

## Supporting information

**S1 Table.**
(DOCX)

## Acknowledgments

The authors wish to thank the editors and reviewers who generously contributed their time and effort to this study.

## Author Contributions

**Conceptualization:** Fengju Wu, Bao Zhu, Siqi Tao.

**Data curation:** Fengju Wu, Siqi Tao.

**Formal analysis:** Fengju Wu, Bao Zhu.

**Methodology:** Fengju Wu.

**Project administration:** Fengju Wu.

**Resources:** Fengju Wu, Bao Zhu.

**Software:** Bao Zhu.

**Writing – original draft:** Fengju Wu.

**Writing – review & editing:** Fengju Wu, Bao Zhu, Siqi Tao.

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
