## [Decision Letter · Decision Letter 0]

18 Dec 2023

PONE-D-23-38843Can good ESG performance of listed companies reduce abnormal stock price fluctuations?--Empirical evidence from ChinaPLOS ONE

Dear Dr. Wu,

Thank you for submitting your manuscript to PLOS ONE. After careful consideration, we feel that it has merit but does not fully meet PLOS ONE’s publication criteria as it currently stands. Therefore, we invite you to submit a revised version of the manuscript that addresses the points raised during the review process.

We look forward to receiving your revised manuscript.

Kind regards,

María del Carmen Valls Martínez, Ph.D.

Academic Editor

PLOS ONE

Journal Requirements:

2. We note that your Data Availability Statement is currently as follows: "All relevant data are within the manuscript and its Supporting Information files".

Reviewers' comments:

Reviewer's Responses to Questions

**Comments to the Author**

1. Is the manuscript technically sound, and do the data support the conclusions?

Reviewer #1: Yes

Reviewer #2: Yes

Reviewer #3: Yes

Reviewer #4: Yes

Reviewer #5: Partly

2. Has the statistical analysis been performed appropriately and rigorously? 

Reviewer #1: Yes

Reviewer #2: Yes

Reviewer #3: Yes

Reviewer #4: Yes

Reviewer #5: Yes

3. Have the authors made all data underlying the findings in their manuscript fully available?

Reviewer #1: Yes

Reviewer #2: Yes

Reviewer #3: Yes

Reviewer #4: Yes

Reviewer #5: Yes

4. Is the manuscript presented in an intelligible fashion and written in standard English?

Reviewer #1: Yes

Reviewer #2: Yes

Reviewer #3: Yes

Reviewer #4: Yes

Reviewer #5: Yes

5. Review Comments to the Author

Reviewer #1: Thank you for allowing me to review the paper entitled "Can good ESG performance of listed companies reduce abnormal stock price fluctuations?-Empirical evidence from China" Here are some suggestions to improve the paper:

1. Originality: Does the paper contain new and significant information adequate to justify publication?

1. It is a good initiative by the authors which to examine the ESG performance of listed companies with stock price which is one of the current issues. However, the intention of this study is not clearly outline. Author should explain and discuss the so what question. What is the urgency of this study?

2. Next, please make a stronger case as to why your paper is needed and how you contribute to current academic discussion. Although authors have presented the gap in the introduction, but it is not sufficient. You should be more convincing and using the current literature review to indicate the gap that needs to be closed. You have to explain why the gap matters. Why is the gap a real concern? What will go wrong if the gap is unaddressed? The authors need to explain better what is the main puzzle their research is addressing.

3. Authors has highlighted one past empirical study. However, authors should provide more elaboration and highlight deficiencies or gaps in the existing literature. The authors should offer a more comprehensive analysis of the limitations present in previous studies. By doing so, author can provide valuable insights into how your own research diverges from and addresses those gaps. This will help establish the novelty and unique contributions of their study, especially the theoretical aspect.

4. I would like to suggest author to revise the introduction which can consider following flow of the introduction and addressing the following questions: (1) Briefly describe and illustrate the current issue. (2) Why is such a study with proposed research gaps important? (3) How does this research gap relate to the current issue? (4) Why is such an underexplored piece of work important to be tested in your study? (5) Are there any similar studies conducted in the past? (6) What is the uniqueness of this study compared to past empirical studies? (7) What are the research objectives? (8) What are the contributions of the study?

2. Relationship to Literature: Does the paper demonstrate an adequate understanding of the relevant literature in the field and cite an appropriate range of literature sources? Is any significant work ignored?

5. Literature review is required. Authors should bear in mind that a good literature review is a comprehensive and well-structured synthesis of existing scholarly works and research findings on a specific topic. It involves identifying, analyzing, and evaluating relevant sources to provide a critical overview of the current knowledge and gaps in the field. Author should provide a critical and comprehensive review, before embarking to any study method.

6. Hypotheses can be more straightforward, especially the mediating path.

3. Methodology: Is the paper's argument built on an appropriate base of theory, concepts, or other ideas? Has the research or equivalent intellectual work on which the paper is based been well designed? Are the methods employed appropriate?

7. I am suggesting authors to revise the title for definitions of variables to operationalization of the variables.

8. Including control variables in the study is indeed a good practice as they can help account for potential confounding factors that may influence the results. To improve the paper, it is important to include control variables and provide reasonable justifications for their inclusion.

9. The authors should discuss the generalizability and representativeness of their sample in relation to the target population. Authors need to clearly explain how the chosen sample is intended to be representative of and reflective of the larger population. Any strategies employed to ensure a diverse and inclusive sample should also be highlighted. This will increase the credibility of the research findings and help readers understand the extent to which the results can be generalized to the broader population.

10. Is this data sufficient to draw a robust finding? Why was this secondary data used? Any other types of data could be used?

11. The authors should provide adequate justification for the use of the software in this study. They should explain why this particular software was chosen and how it aligns with the research objectives. By elaborating on the software's specific functionalities and how it supports the analysis of the data, the authors can enhance the validity and reliability of their findings.

5. Practicality and/or Research implications: Does the paper identify clearly any implications for practice and/or further research? Are these implications consistent with the findings and conclusions of the

12. The ending part of the manuscript is not well written.

13. I would suggest author to structure their conclusion part of the manuscript as follows: (1) Discussion, (2) theoretical Implications, (3) practical implications/policy implications, (4) Limitations and Future Research Recommendations, and (5) Conclusion.

14. Should have a standalone section for discussion, the authors need to ensure that the key findings or scale development are discussed. The discussion section is where you delve into the meaning, importance, and relevance of your results. It should focus on explaining and evaluating what you found, showing how it relates to your literature review and research questions, and making an argument in support of your overall conclusion. Limited citations. Please cite more to support the discussion.

15. For theoretical implications. How can you imply from the findings? This section should discuss the implications of the study's findings and how they contribute to the existing theoretical knowledge. Summarize the key findings and their relevance to the existing theoretical frameworks or models. Analyze how the findings align with or challenge current theoretical perspectives and concepts related to all the key concepts of this study. Discuss any theoretical insights or advancements that the study provides and highlight how the findings contribute to a deeper understanding of the research area.

16. Should have a standalone section for practical implications: the authors should provide valuable insights based on current practices and policies, supported by evidence from their research. To strengthen the practical implications, it is crucial to reference specific findings, data, or examples that demonstrate the validity and reliability of the recommendations. By incorporating this approach, the authors can offer concrete and actionable suggestions that have a solid grounding in their research findings.

17. Standalone section for limitation and future research recommendation, it is not sufficiently written. Please reconsider and identify limitations of the study.

6. Quality of Communication: Does the paper clearly express its case, measured against the technical language of the field and the expected knowledge of the journal's readership? Has attention been paid to the clarity of expression and readability, such as sentence structure, jargon use, acronyms, etc.

Proofreading is required. Cite more recent studies. Relook into the structure of the manuscript.

Reviewer #2: Review Report for Manuscript ID: PONE-D-23-38843. “Can good ESG performance of listed companies reduce abnormal stock price fluctuations?--Empirical evidence from China” I applaud the authors for their efforts in crafting this article. However, there are a number of issues with this paper that need to be revised. These concerns are delineated as follows:

1 In line 345, there is a problem with the three-line table setting in Table 2.

2 In line 302, there are problems with literature citations, and the same problem also exists in many places.

3 In lines 323 – 332, the authors do not explain in detail the meaning of the parts of the formula.

4 Suppose a problem with H2 setup. This article focuses on the relationship between ESG performance and abnormal stock price fluctuations. Why do we still need to explore the relationship between ESG performance and investor attention? The authors set investor attention as an intermediary variable, but should also combine H2 and H3.

5 In the heterogeneity analysis section, why is the total sample volume of the enterprise life cycle and the bull and bear market inconsistent?

6 In the literature review section, the author did not do a good job of literature together, many places directly put the literature, so we need to carry out some sorting.

My Recommendation: Major Revision

Reviewer #3: Incorporate the suggested comments in the file attached. You can slight look for new literature in your perspective. There is very good research on the emerging economies and look for the covid-19 based papers they will be good ground as well.

Reviewer #4: For the "Can good ESG performance of listed companies reduce abnormal stock price fluctuations? Empirical evidence from China" PONE-D-23-38843, please see the attachment. please see the attachment.

Reviewer #5: Thanks for the submission.

While I applaud the authors' efforts, the manuscript does not contribute novelty in the research findings.

The manuscript's contribution is already existed in latest published research.

6. PLOS authors have the option to publish the peer review history of their article (what does this mean?). If published, this will include your full peer review and any attached files.

Reviewer #1: No

Reviewer #2: No

Reviewer #3: **Yes: **Shujahat Ali

Reviewer #4: No

Reviewer #5: No

---

## [Author Response · Author response to Decision Letter 0]

21 Jan 2024

Responses to each of the comments made by the academic editors and reviewers have been placed in an attachment with the file name labeled "Response to Reviewers" and have been uploaded separately.

---

## [Decision Letter · Decision Letter 1]

7 Feb 2024

PONE-D-23-38843R1Can good ESG performance of listed companies reduce abnormal stock price fluctuations?----Mediation effects based on investor attentionPLOS ONE

Dear Dr. Wu,

Thank you for submitting your manuscript to PLOS ONE. After careful consideration, we feel that it has merit but does not fully meet PLOS ONE’s publication criteria as it currently stands. Therefore, we invite you to submit a revised version of the manuscript that addresses the points raised during the review process.

We look forward to receiving your revised manuscript.

Kind regards,

María del Carmen Valls Martínez, Ph.D.

Academic Editor

PLOS ONE

Reviewers' comments:

Reviewer's Responses to Questions

**Comments to the Author**

1. If the authors have adequately addressed your comments raised in a previous round of review and you feel that this manuscript is now acceptable for publication, you may indicate that here to bypass the “Comments to the Author” section, enter your conflict of interest statement in the “Confidential to Editor” section, and submit your "Accept" recommendation.

Reviewer #1: All comments have been addressed

Reviewer #2: (No Response)

Reviewer #4: (No Response)

2. Is the manuscript technically sound, and do the data support the conclusions?

Reviewer #1: Yes

Reviewer #2: Yes

Reviewer #4: Partly

3. Has the statistical analysis been performed appropriately and rigorously? 

Reviewer #1: Yes

Reviewer #2: Yes

Reviewer #4: N/A

4. Have the authors made all data underlying the findings in their manuscript fully available?

Reviewer #1: Yes

Reviewer #2: No

Reviewer #4: No

5. Is the manuscript presented in an intelligible fashion and written in standard English?

Reviewer #1: Yes

Reviewer #2: (No Response)

Reviewer #4: No

6. Review Comments to the Author

Reviewer #1: Thank you for allowing me to review the improved version of the manuscript. All my comments are well addressed by the authors. Thus, I have no further comments.

Reviewer #2: (No Response)

Reviewer #4: (No Response)

7. PLOS authors have the option to publish the peer review history of their article (what does this mean?). If published, this will include your full peer review and any attached files.

Reviewer #1: No

Reviewer #2: No

Reviewer #4: No

---

## [Author Response · Author response to Decision Letter 1]

19 Feb 2024

The responses to editor and reviewer are all in a separate attachment, named 'Response to Reviewers-2'

---

## [Decision Letter · Decision Letter 2]

15 May 2024

PONE-D-23-38843R2Can good ESG performance of listed companies reduce abnormal stock price fluctuations?Mediation effects based on investor attentionPLOS ONE

Dear Dr. Wu,

Thank you for submitting your manuscript to PLOS ONE. After careful consideration, we feel that it has merit but does not fully meet PLOS ONE’s publication criteria as it currently stands. Therefore, we invite you to submit a revised version of the manuscript that addresses the points raised during the review process.

We look forward to receiving your revised manuscript.

Kind regards,

María del Carmen Valls Martínez, Ph.D.

Academic Editor

PLOS ONE

Reviewers' comments:

Reviewer's Responses to Questions

**Comments to the Author**

1. If the authors have adequately addressed your comments raised in a previous round of review and you feel that this manuscript is now acceptable for publication, you may indicate that here to bypass the “Comments to the Author” section, enter your conflict of interest statement in the “Confidential to Editor” section, and submit your "Accept" recommendation.

Reviewer #6: (No Response)

Reviewer #7: (No Response)

2. Is the manuscript technically sound, and do the data support the conclusions?

Reviewer #6: Yes

Reviewer #7: Partly

3. Has the statistical analysis been performed appropriately and rigorously? 

Reviewer #6: Yes

Reviewer #7: No

4. Have the authors made all data underlying the findings in their manuscript fully available?

Reviewer #6: Yes

Reviewer #7: No

5. Is the manuscript presented in an intelligible fashion and written in standard English?

Reviewer #6: Yes

Reviewer #7: No

6. Review Comments to the Author

Reviewer #6: Line 244: “information” should be “Information”

Line 272 “Sample selection and data sources” the reasons to excluding the three types of companies as a sample should be explained

Line 311: put a space between al. and reference, and also between reference and “to” (Zhang et al.[30]to)

Line 389: please give a space between Jiang and reference number - Jiang[32]

Line 476: after reference number should be “.” Instead of “,” and give a space between “.” and “If” (s [36],If)

Line 535: in the “Recommendation” section it would be better to include the research limitation and future research in relation to the ESG concept

Reviewer #7: I have read the study and found that the manuscript has some merits. However, I have some comments:

1. The context of research is ommited in the introduction section. The motivation and the contribution of the study are presented, but the readers can not find that they were be supported by privious studies.

2. Theorization is average, the literature review section were presented by listing. The backgroud literature is missing. The readers will question whether the authors are based on what literature or theories to propose variables as Equation 1,2,3. While carrying out the review of literature, the authors should break the review into different themes, based on their choice of variables. It will not only make the review of literature more systematic, but also will provide a rationale behind choice of the variables. Moreover, the literature review should provide a basis for the study by substantiating the research gap.

3. The authors used several approaches. However, please discuss why them is adopted? What are the conditions and advantages of them?

4. The Pooled OLS and GMM approaches have some limitations. In the “Empirical results” section, please provide more diagnostic tests, such as slope heterogeneity test. I think there might be a problem with slope heterogeneity among the sampled countries. Please show advantages of these approaches for the readers.

5. The policy recommendations are weak and sound more generic. Authors need to provide policy recommendations in accordance with the results. Can you please be direct about what policies that administrators in China should be implemented? And, why?

7. PLOS authors have the option to publish the peer review history of their article (what does this mean?). If published, this will include your full peer review and any attached files.

Reviewer #6: **Yes: **Nyoman Indah Kusuma Dewi

Reviewer #7: No

---

## [Author Response · Author response to Decision Letter 2]

19 Jun 2024

PONE-D-23-38843R3

Can good ESG performance of listed companies reduce abnormal stock price volatility? Mediating Effects Based on Investor Attention

PLOS ONE

Dear Dr. Valls Martínez,

We would like to submit our revised manuscript to PLOS ONE.We sincerely thank you and the reviewers for your valuable comments and suggestions, which have greatly improved the quality of our manuscript. All of us authors have carefully read the comments , and have discussed and revised each of these issues. Furthermore, we have carefully revised and polished the entire manuscript to enhance its readability. All changes are marked in red. 

We believe that these revisions have greatly improved the quality of the manuscript. Thank you for giving us this opportunity to revise our paper based on the reviewers' valuable comments. We hope that our revisions meet the reviewers' expectations and look forward to hearing from you.

We have tried our best to make all the revisions clear, and we hope that the revised manuscript can satisfy the requirements for publication. Below are our main responses to the reviewers' comments.

Response to the comments of

Reviewer #6 

Comment 1: Line 244: “information” should be “Information”

Line 311: put a space between al. and reference, and also between reference and “to” (Zhang et al.[30]to)

Line 389: please give a space between Jiang and reference number - Jiang[32]

Line 476: after reference number should be “.” Instead of “,” and give a space between “.” and “If” (s [36],If)

Response: Thank you for your meticulous review. The formatting issues you identified, which were present in the first draft, have all been addressed. Furthermore, we have thoroughly reviewed the entire manuscript and corrected any grammatical and formatting errors to ensure it conforms to the journal's style guidelines.

Comment 2: Line 272 “Sample selection and data sources” the reasons to excluding the three types of companies as a sample should be explained.

Response: Thank you for your insightful comments. In response to your request for clarification on the exclusion criteria for certain company types, we have now detailed the reasons for excluding the following categories of companies in the revised manuscript:

Companies in special industries such as finance and insurance are excluded due to their distinct business models and risk profiles, which may compromise the generalizability of our findings.

Companies with incomplete data for key variables are excluded to avoid biases that could distort the research outcomes.

Companies with special designations like ST and *ST, as well as those with abnormal operations, are excluded because their financial data and stock price volatility are significantly different from those of companies operating under normal conditions.

We appreciate the opportunity to refine our manuscript based on your feedback and hope that our revisions meet with your approval.

Comment 3: Line 535: in the “Recommendation” section it would be better to include the research limitation and future research in relation to the ESG concept.

Response: Thank you for your valuable feedback. In response to your suggestion, we have incorporated a new section, "7.3 Research Limitations and Future Research Prospects," to address the limitations and potential avenues for future work.

We have acknowledged the lack of a unified ESG evaluation system and its implications for our study's findings. Additionally, we recognize the limitations in measuring investor attention using the Baidu index and suggest that more precise metrics are needed.

Looking ahead, we propose that future research should aim to develop a robust ESG rating framework and explore more accurate methods for gauging investor attention. We also recommend examining the impact of ESG performance on stock price volatility across various industries and regions, as well as the influence of individual ESG components.

We trust that these revisions provide a clearer understanding of our study's scope and limitations, and we appreciate the opportunity to enhance our manuscript based on your guidance.

Response to the comments of

Reviewer #7 

 Comment 1：The context of research is ommited in the introduction section. The motivation and the contribution of the study are presented, but the readers can not find that they were be supported by privious studies.

Response: Thank you for your insightful feedback. We have revised the introduction to include a theoretical framework and a concise literature review that support our research context and contributions.

In the second paragraph, we've incorporated risk management theories to elucidate the risk reduction effects of ESG performance across subcomponents. We've also provided a brief review of the literature on ESG's impact on abnormal stock price volatility, noting the existing lack of consensus.

The third paragraph now includes insights from signaling and reputation theories, underscoring the existing literature's findings on the link between ESG performance and investor attention, and the potential mediating role of investor attention in the relationship between ESG performance and stock price volatility. We've identified a gap in the literature regarding this mediating mechanism.

The revised introduction is now grounded in theory and supported by literature. We hope these revisions meet your expectations.

Comment 2：Theorization is average, the literature review section were presented by listing. The backgroud literature is missing. The readers will question whether the authors are based on what literature or theories to propose variables as Equation 1,2,3. While carrying out the review of literature, the authors should break the review into different themes, based on their choice of variables. It will not only make the review of literature more systematic, but also will provide a rationale behind choice of the variables. Moreover, the literature review should provide a basis for the study by substantiating the research gap.

Response: Thank you for your insightful suggestions. We have thoroughly revised the literature review section, organizing it thematically around our research hypotheses and variable selection to provide a clear theoretical basis for our study.

We have removed the original literature review on the economic consequences of ESG performance, as it did not directly relate to our research hypotheses. Instead, we have focused the review on the relationship between a firm's ESG performance and abnormal stock price volatility, as well as the impact of investor concerns on stock price volatility. This thematic approach not only systematizes the literature review but also directly supports the rationale behind our variable selection.

Our review highlights the lack of consensus in the existing literature on the effects of ESG performance on stock price volatility and identifies a significant gap regarding the mediating role of investor attention. This study aims to address these gaps, and the revised literature review now serves as a robust foundation for our research, substantiating the research gap and supporting our theoretical framework.

We trust that these enhancements meet your expectations and provide a solid basis for our study.

Comment 3：The authors used several approaches. However, please discuss why them is adopted? What are the conditions and advantages of them?

Response: Thank you for your constructive suggestions. We have carefully considered your feedback regarding the use of various methods for robustness and endogeneity tests in our study. In the revised manuscript, we have provided a detailed discussion on the rationale behind the selection of each method, along with their respective conditions and advantages. Specifically, we have:

Addressed the lack of standardized ESG ratings by replacing the explanatory variables to ensure the robustness of our results. This approach acknowledges the variability in evaluation criteria and coverage among rating agencies.

Employed the Generalized Method of Moments (GMM) estimation to verify the robustness of our dynamic panel data model, which is particularly advantageous for dealing with potential issues of endogeneity inherent in such models.

Utilized Propensity Score Matching (PSM) to mitigate endogeneity arising from sample selection bias, recognizing that a firm's ESG performance may be influenced by its financial status and governance.

When we use each method in the manuscript, we briefly describe the reasons, conditions, and advantages of using that method.We trust that these revisions provide a comprehensive understanding of our methodological choices and their relevance to our study.

Comment 4：The Pooled OLS and GMM approaches have some limitations. In the “Empirical results” section, please provide more diagnostic tests, such as slope heterogeneity test. I think there might be a problem with slope heterogeneity among the sampled countries. Please show advantages of these approaches for the readers. 

Response: Thank you for your meticulous review and valuable suggestions. Your comments on the limitations of the Pooled OLS and GMM methods are very pertinent and are essential to improve the quality and credibility of our study.

In response to your proposal for a slope heterogeneity test, we have given it in-depth consideration and discussion. We recognize that the slope heterogeneity test is an important tool for diagnosing potential nonlinearities in the model or differences between different subsamples. However, in the current study, given the large differences in economic structures and policy environments between our samples, we believe that the slope heterogeneity test may not adequately capture these complex dynamics. Therefore, we opted for a diagnostic test that is more appropriate to the context of our study.

To enhance the robustness of our results, we add the following diagnostic tests in the "Empirical results" section:(1) Propensity Score Matching (PSM): We used PSM to address the issue of potential selection bias by ensuring that the treatment and control groups were similar on observable characteristics prior to treatment. (2) Instrumental variables approach: we chose appropriate instrumental variables for two-stage least squares regression analyses to address possible endogeneity issues.

In addition, we agree that the reader needs to be aware of the advantages of the methods we employ. In the revised manuscript, we describe the advantages of these methods.By adding more diagnostic tests and elaborating on the advantages of these methods, our study is more comprehensive and reliable.

Comment 5：The policy recommendations are weak and sound more generic. Authors need to provide policy recommendations in accordance with the results. Can you please be direct about what policies that administrators in China should be implemented? And, why?

Response: Thank you for your insightful comments. We have revised our policy recommendations to include three actionable measures for businesses, investors and government agencies. In particular, the recommendations for government agencies detail the specific policies that should be implemented and why. This includes three main points:

(1) Provide ESG financial incentives: We recommend that the government provide subsidies and tax incentives to encourage listed companies to enhance their environmental and social responsibilities and improve the ESG disclosure system. (2) Enhanced environmental regulation: We suggest stricter pollution control through an emissions licensing system, tougher penalties for "greenwashing", and incentives for compliance through market mechanisms such as emissions charges. (3) Enhance ESG Information Transparency: We recommend mandatory and free public disclosure of ESG ratings to ensure that investors have access to accurate information, reduce stock price volatility, and support capital market stability.

These recommendations are based on the results of our empirical research, which emphasizes the significant impact of corporate ESG performance on stock price volatility and investor decision-making. We have ensured that the revised manuscript clearly articulates the rationale behind each of the policy recommendations and how they can be effectively implemented.

Sincerely,

Fengju Wu

E-mail: wufengju@naujsc.edu.cn

---

## [Decision Letter · Decision Letter 3]

7 Jul 2024

Can good ESG performance of listed companies reduce abnormal stock price volatility? Mediation effects based on investor attention

PONE-D-23-38843R3

Dear Dr. Fengju Wu,

We’re pleased to inform you that your manuscript has been judged scientifically suitable for publication and will be formally accepted for publication once it meets all outstanding technical requirements.

Kind regards,

María del Carmen Valls Martínez, Ph.D.

Academic Editor

PLOS ONE

Reviewer's Responses to Questions

**Comments to the Author**

1. If the authors have adequately addressed your comments raised in a previous round of review and you feel that this manuscript is now acceptable for publication, you may indicate that here to bypass the “Comments to the Author” section, enter your conflict of interest statement in the “Confidential to Editor” section, and submit your "Accept" recommendation.

Reviewer #6: (No Response)

Reviewer #7: All comments have been addressed

2. Is the manuscript technically sound, and do the data support the conclusions?

Reviewer #6: (No Response)

Reviewer #7: Yes

3. Has the statistical analysis been performed appropriately and rigorously? 

Reviewer #6: (No Response)

Reviewer #7: Yes

4. Have the authors made all data underlying the findings in their manuscript fully available?

Reviewer #6: (No Response)

Reviewer #7: Yes

5. Is the manuscript presented in an intelligible fashion and written in standard English?

Reviewer #6: (No Response)

Reviewer #7: No

6. Review Comments to the Author

Reviewer #6: (No Response)

Reviewer #7: The effort of the authors in the revised manuscript is appreciated. All my recommendations are addressed. Big congrats.

7. PLOS authors have the option to publish the peer review history of their article (what does this mean?). If published, this will include your full peer review and any attached files.

Reviewer #6: **Yes: **Nyoman Indah Kusuma Dewi

Reviewer #7: No

---

## [Editor Report · Acceptance letter]

26 Jul 2024

PONE-D-23-38843R3 

PLOS ONE

Dear Dr. Wu, 

I'm pleased to inform you that your manuscript has been deemed suitable for publication in PLOS ONE. Congratulations! Your manuscript is now being handed over to our production team.

Kind regards, 

on behalf of

Dr. María del Carmen Valls Martínez 

Academic Editor

PLOS ONE